# Integrating Ecological Knowledge into Regenerative Design: A Rapid Practice Review

Jane Toner *, Cheryl Desha, Kimberley Reis, Dominique Hes and Samantha Hayes

Cities Research Institute, Griffith University, Brisbane 4111, Australia; c.desha@griffith.edu.au (C.D.); k.reis@griffith.edu.au (K.R.); d.hes@griffith.edu.au (D.H.); s.hayes@griffith.edu.au (S.H.)
* Correspondence: jane.toner@griffithuni.edu.au

**Abstract:** While sustainable design practice is working to reduce the ecological impacts of development, many of the earth's already damaged life support systems require repair and regeneration. Regenerative design theory embraces this challenge using an ecological worldview that recognizes all life as intertwined and interdependent to deliver restorative outcomes that heal. Central to regenerative design theory is the mutually beneficial and coevolving 'stewardship' relationship between community and place, the success of which requires local ecological knowledge. However, there is a lack of understanding about how—within the design process—practitioners are integrating 'innate knowledge' of place held by local people. This rapid practice review sought to collate and evaluate current 'regenerative design practice' methods towards ensuring good practice in the integration of place-based ecological knowledge. A comprehensive online search retrieved 345 related articles from the grey literature, academic book chapters, and government reports, from which 83 articles were analyzed. The authors conclude that regenerative design practice is emergent, with the design practice of including community knowledge of ecological systems of place remaining ad hoc, highly variable, and champion-based. The findings have immediate implications for regenerative design practitioners, researchers, and developers, documenting the state of progress in methods that explore innate ecological knowledge and foster co-evolving ecological stewardship.

**Keywords:** regenerative design; place-based; built environment; urban ecology; ecological wisdom; living systems; co-evolution; rapid practice review

## 1. Introduction

Human activity has dramatically shaped Earth's physical and living systems, impacting the viability of many planetary systems [1–3]. In particular, the built environment has caused severe biological diversity loss, disturbing and fragmenting ecosystems [4], creating pollution [5,6], and altering atmospheric conditions [7]. Urban development continues to promote the separation of humanity from non-human life [8–12], reflecting dominant socio-cultural paradigms derived from Eurocentric, 'mechanistic' perspectives [9,13]. Many critics argue that mechanistic perspectives have also led to improvement strategies that only incrementally reduce harmful practices, without addressing the damage that has been done [13–16].

Recognizing the interdependence between humanity and nature is crucial to address these pressing planetary health challenges [17,18]. Built environment practitioners have been working to reduce the environmental impacts of cities around the world, motivated by calls to action from international agencies such as the United Nations (UN), World Green Building Council (WGBC), and International Union of Architects (UIA) [12,19,20]. Global initiatives have provided directions and goals to support practitioners, including the Millennium Goals (MDGs) and the Sustainable Development Goals (SDGs) [5,21–23]. However, for some, the idea of a 'sustainable city' is a contradiction of terms because, as

suggested by Rees (1997), cities need to draw on resources from ecosystems much larger than their own to be 'sustained' [8] (p. 305).

Counter to the mechanistic worldview, 'regenerative design and development' embraces an ecological worldview where Earth is acknowledged as a complex, adaptive, and dynamic living system. Within the context of the built environment addressing humanity's biological and social needs [8,24], the ethos of regenerative design and development is to restore and foster new relationships between humans and natural systems so that all life might coevolve and thrive [14,15]. From this ecological perspective, humans and human systems are entwined within interconnected and interdependent living systems [25–27]. Several authors see sustainability and regenerative paradigms as a precursor to transforming the mechanistic worldview [15,26,28]. This shift in perceptual insight would allow cities, buildings, and their supporting infrastructures to be viewed as inherently regenerative living systems [25,29–31], with place-based attuned decision making about design and development [32].

Key to regenerative design and development is a practitioner shift in mindset from a mechanistic worldview to an ecological worldview, moving from reductionist assumptions to systems thinking [15,26,33,34]. Familiarity with ecological knowledges and systems thinking would enable practitioners to integrate the principles of living systems into practice [30,34], however, formal education and professional development are currently lacking in this field [25]. Within this context, this research explored how place-based ecological knowledge is being integrated into regenerative design paradigms within the built environment. This paper presents the results of a rapid practice review of the literature exploring the strategies for integrating ecological knowledge into regenerative design practice. It establishes a broad foundation for future research into the benefits of enhancing the ecological knowledge of built environment professionals and communities. These benefits potentially include shifting mindsets towards an ecological worldview, empowering communities to act as stewards for their local ecosystems and reshaping the built environment towards net positive outcomes to holistically realize the SDGs.

## 2. Methods

A rapid review method was used to survey the literature and obtain broad insights into how ecological knowledge is integrated into sustainable and regenerative design projects in the built environment. Rapid reviews are streamlined systematic literature reviews that expedite the collection and synthesis of evidence-based research [35]. They are increasingly seen as an appropriate method for synthesizing existing research to generate knowledge and inform policy development for the built environment [36,37]. The rapid review method was considered appropriate for this research because regenerative design and development practice is an emergent topic, with progress being relatively recent and rapid, and documented in practice-based publications. Drawn from the protocol outlined by Lagisz et al. [37], this rapid review followed the five (5) steps suggested by Eon et al. [36], as detailed in the following sub-sections. The emergent nature of this method allowed the authors to explore novel approaches within the screening process while still meeting the requirements for rigor and non-bias.

### 2.1. Defining the Problem for Rapid Review

Rapid reviews are often conducted for a specific stakeholder group who are included in iterative discussions throughout the review process to ensure their needs are met and to improve the emerging responses to the review questions [38]. Due to the breadth of knowledge, skills and industry experience of the authorship team, the problem to be addressed could be confidently defined. The overarching research question explored through the literature review was, 'How is place-based ecological knowledge integrated into regenerative design and development projects in the built environment?' This question was addressed by posing four sub-questions, as illustrated schematically in Figure 1.

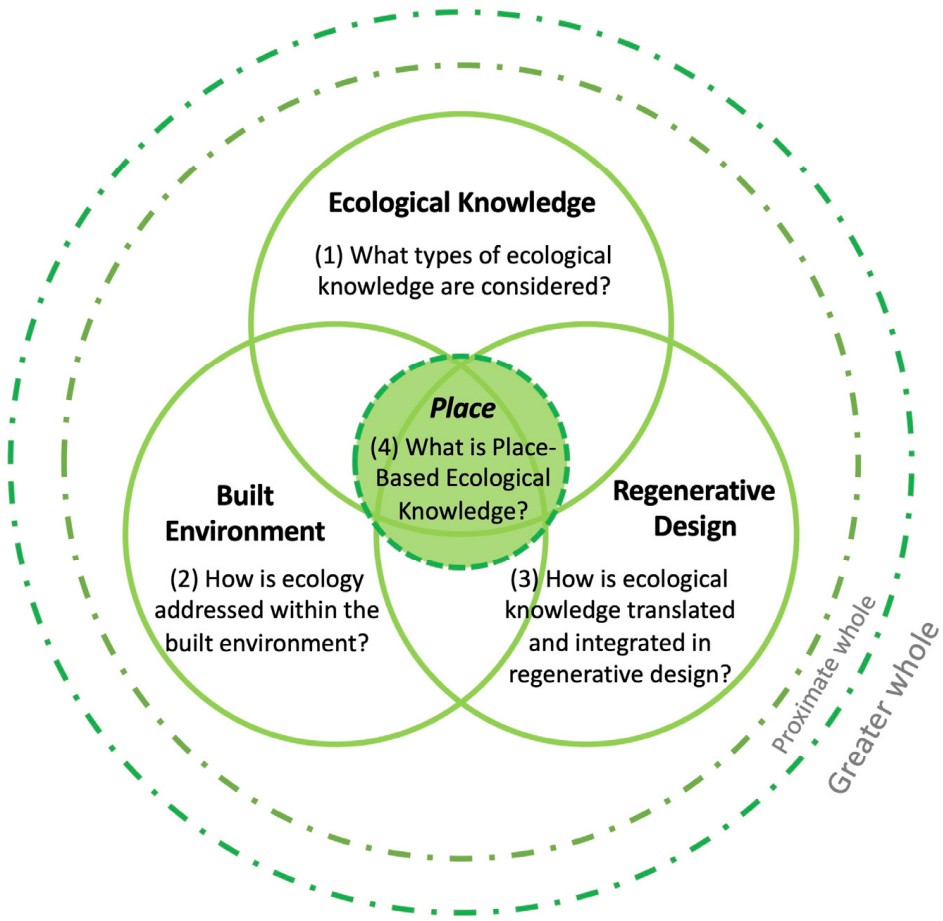

**Figure 1.** Illustrative schematic of the keywords used in the search strategy and their relationship to each other when asking the four research sub-questions shown. (Source: Authors).

This illustration helps to visualize the relationships among the four domains of inquiry, including: (1) the types of ecological information used to inform design; (2) how ecology is addressed in the built environment sector; (3) how ecological knowledge is translated and integrated into regenerative design and development processes, and (4) what is place-based ecological knowledge. The central circle denotes the importance of 'place' as a connected context for each keyword in the sub-questions. Drawing on the language introduced by the Regenesis Group, the circles surrounding the Venn diagram represent the greater systems within which this research is embedded [25] (p. 46). These systems include the 'proximate' whole (i.e., the practitioner context of built environment professionals and communities of practice) and the 'greater' whole (the institutional and statutory context that could be positively impacted through this research).

### 2.2. Search String and Filtering Process

The search strategy included different synonyms and combinations of keywords relevant to regenerative design, ecological knowledge, and the built environment, as shown in Table 1. The terms and keywords in the columns were combined into search strings using the 'OR' search strategy, while the terms and keywords in the rows were combined using 'AND' combinations.

**Table 1.** Key terms used for rapid review database searches.

| Key Terms | Regenerative~ OR Sustainable~ | Ecological Knowledge | Built Environment |
|---|---|---|---|
| Search string terms | Design Development Practice Architecture Thinking Place | Ecoliteracy Eco-literacy Ecological~ Knowledge Understanding Awareness Wisdom Literacy | Urban environment Urban planning Site planning Site analysis |

This rapid practice review used a truncated systematic literature review methodology to identify relevant articles, minimize risk, and reduce biases [36]. Academic papers, as artefacts of evidence, were sought in search-engine databases that are multidisciplinary and related to the built environment. The databases included Scopus, Web of Science, ProQuest (Design and Applied Art Index, Education, ERIC), and EBSCO Host (GreenFILE). The search yielded 126 documents that were imported into Covidence, a web-based screening and data extraction platform.

The research team noted that many well-known 'practice-led' regenerative design researchers (i.e., practitioners who write about the topic) were not represented in the primary database search. A Google Scholar search using the same search terms was subsequently undertaken to supplement the database articles with additional academic articles and grey literature, such as book chapters, reports, commentaries, and practice notes. This search yielded 238 articles, which included many by influential and well-cited authors in the field of regenerative design theory. From the combined total of 364 documents, Covidence detected 24 duplicates, with a further 10 duplicates identified manually. Excluding these from the body of literature identified 330 documents for title and abstract screening against the pre-defined eligibility criteria outlined in Table 2.

**Table 2.** Title and abstract screening pre-defined eligibility criteria.

| Inclusion Criteria | Exclusion Criteria |
|---|---|
| <ul><li>English language.</li><li>Published within last 25 years (1997–2022).</li><li>Discusses how ecological knowledge is gathered and sourced for sustainable and regenerative design and development projects.</li><li>Discusses processes and strategies by which ecological knowledge is integrated into design processes.</li><li>Relates to the built environment, urban environment, urban planning, architectural design and development, or biomimicry.</li></ul> | <ul><li>Languages other than English.</li><li>Topics primarily focused on agriculture, biology, chemistry, economics, fishing, forestry, health and medicine, and structural engineering.</li><li>Poor-quality articles.</li><li>Not easily accessible articles (e.g., behind a paywall, a whole book).</li></ul> |

### 2.3. Document Screening

Two members of the authorship team independently screened the titles and abstracts of all 330 articles using Covidence, a platform designed to manage the process of identifying relevant studies for a systematic review. Conflicts were resolved through further discussion within the authorship team. Through this process, 126 documents were deemed irrelevant and dismissed. This resulted in a body of literature of 204 potentially relevant articles consisting of 76 articles from the primary databases and 128 articles from Google Scholar.

Following this, the remaining articles were considered in three batches, or 'subsets'. The lead author reviewed the full text of articles from the primary databases, which led

to the selection of 30 articles forming 'Subset 1'. Further selection criteria were applied to the Google Scholar articles to streamline the screening process while meeting the aims of the rapid practice review. Articles from Google Scholar from 2018 to 2022 were screened to capture information on the most recent developments in the field, forming 'Subset 2' with 23 articles. 'Subset 3' comprised 22 articles (also selected from Google Scholar) by recognized experts in 'regenerative design' and 'place-based biomimicry'. The resultant workflow of the screening process is illustrated in the adapted PRISMA diagram shown in Figure 2.

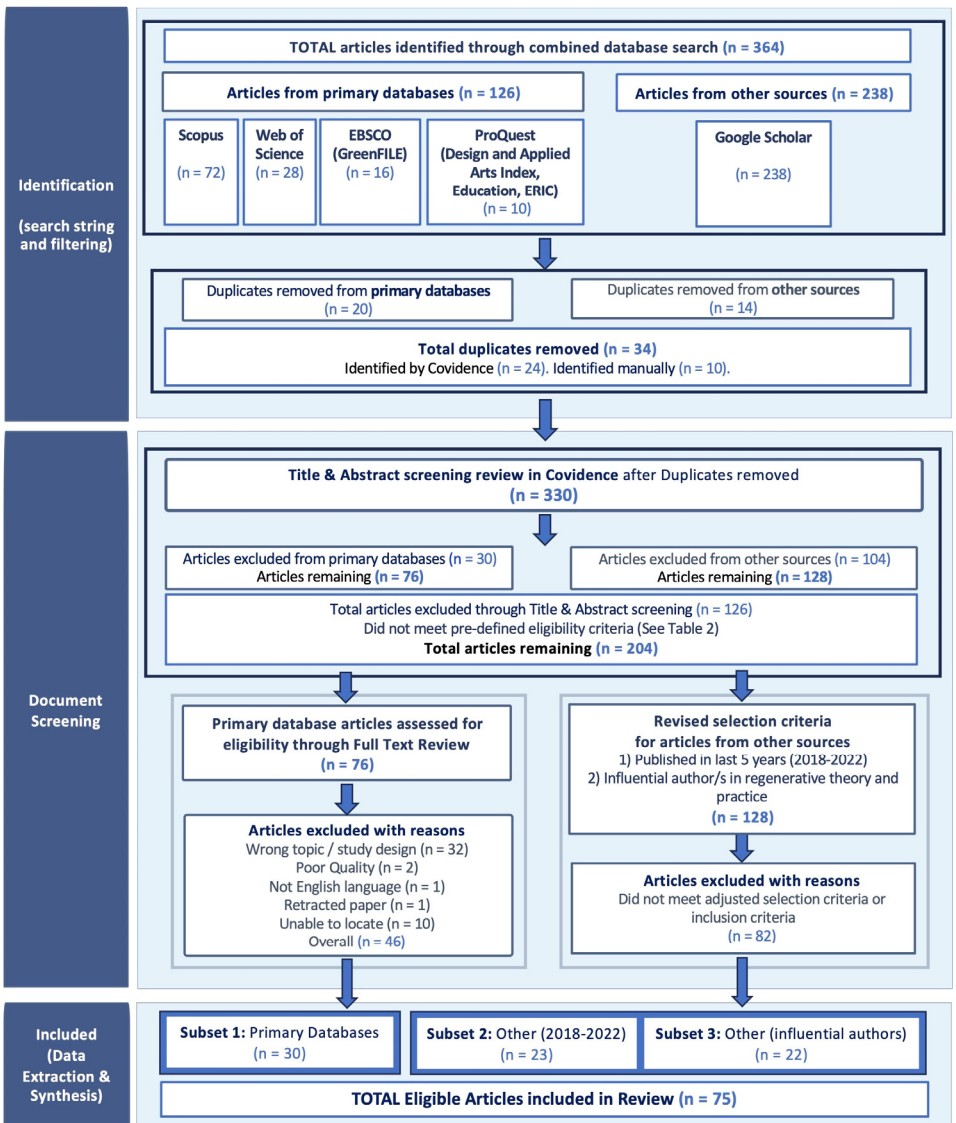

**Figure 2.** PRISMA flow diagram showing the process used to select articles for inclusion in the rapid review process (based on the PRISMA diagram from Covidence).

## *2.4. Data Extraction and Synthesis*

Following the document screening and selection process, the key themes of this review were defined and analyzed using an a priori list of terms that evolved as the review progressed. Data visualization, using Voyant Tools, a web-based software platform (v. 2.6.9) [39], was used to streamline the extraction and synthesis of data across the subsets to inform the descriptive results detailed in Section 3, and to explore the themes detailed in Section 4. Voyant Tools was also used to assess the quality of the overall dataset by comparing the three subsets to confirm the validity of the Google Scholar subsets.

*2.5. Quality Assessment of Selected Articles*

Ensuring the quality of articles selected in rapid reviews is vital to provide confidence in the findings. The process described and illustrated in Figure 2 was intended to reduce the potential systematic bias in the article selection. The elements of this process contributing to the confidence in the reliability of the articles selected included double screening through Covidence and a full text review of the primary database articles. These steps established the validity of 'Subset 1' as a control to assess the two subsets extracted from the Google Scholar articles.

Data visualization is a novel emergent method used to enhance the trustworthiness and dependability of qualitative data by comparing different subsets of data [40]. The three datasets were analyzed with Voyant Tools 'Trends Stacked Bar Charts' by comparing the frequency of eight words of significance to this study, being: regenerative, sustainable, ecological, knowledge, cities, urban, neighborhood, and community. The data visualizations are shown in Figure 3 for the articles found through the primary database search ('Subset 1'), Figure 4 for the additional articles searched for between 2018 and 2022 ('Subset 2'), and Figure 5 for the additional articles searched for by influential regenerative design experts ('Subset 3'). The numbers along the *x*-axis represent the articles within the subset and correspond with a key on the right indicating the article's first author and publication year. The *y*-axis represents the relative frequency of the selected words.

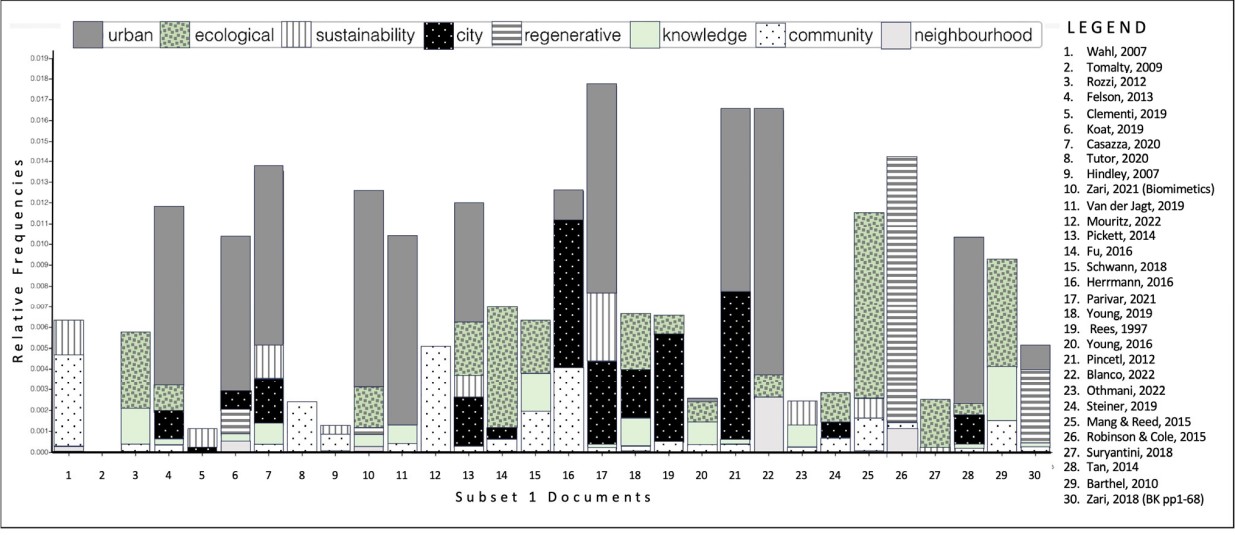

**Figure 3.** Subset 1—primary database articles (consisting of 30 scholarly articles).

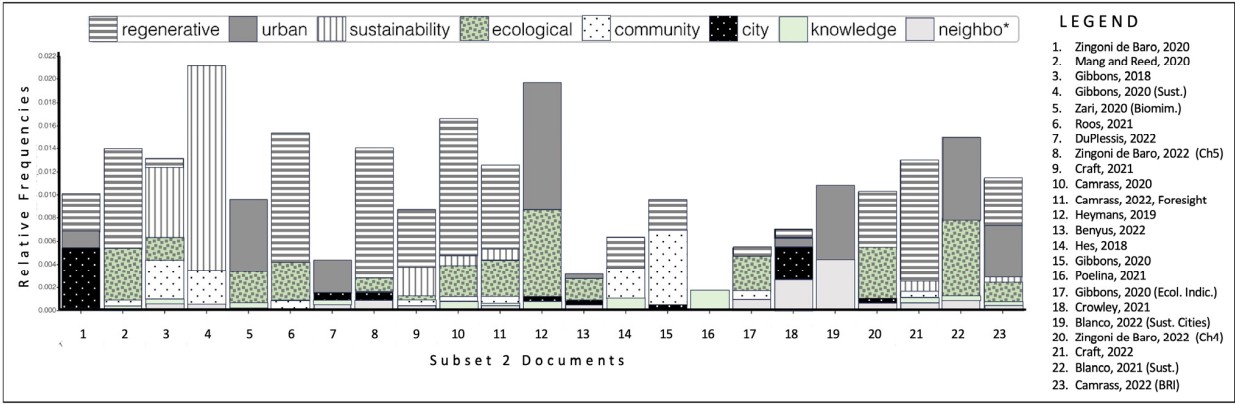

**Figure 4.** Subset 2—Google Scholar 'last 5 years' articles (consisting of 23 academic and grey literature articles, 2018–2022).

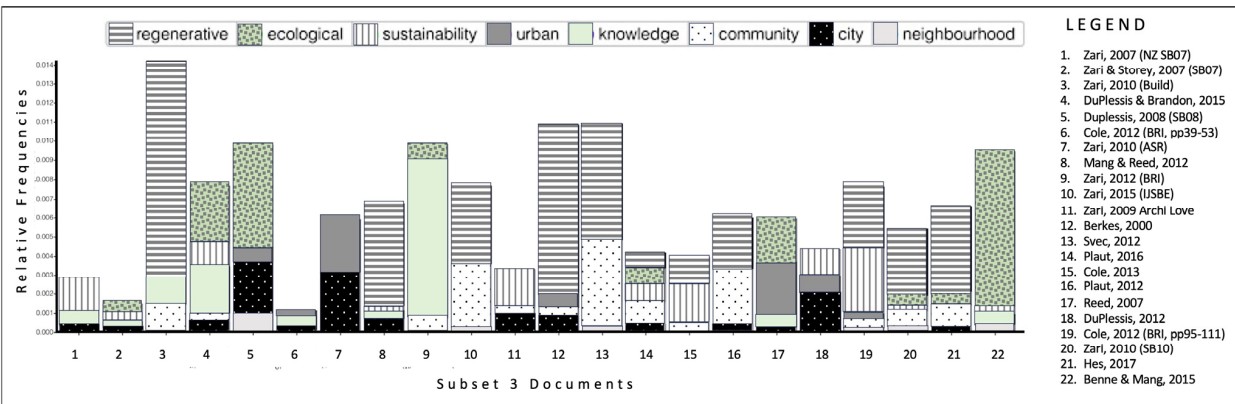

**Figure 5.** Subset 3—Google Scholar 'expert' articles (consisting of 22 academic and grey literature articles).

The overall similarity in the frequency of the significant words and the observed correlations among the three subsets provide confidence that the article selected are representative of the field. 'Urban' was the most frequently used significant word in each dataset. It signifies the dominant scale of the built environment addressed in the full dataset, as it ranks among the top four words for each subset. Urban scale was most frequently referenced in the literature in relation to the disciplines of ecology, design, and planning. The least frequently used significant word to describe scale was 'neighborhood', but it was consistently used across the full dataset. This aligns with the literature observations about the neighborhood scale being the 'fundamental urban design unit' [41]. Both Subsets 2 and 3 contained the same top four significant words: regenerative, urban, sustainable, and ecological.

### 2.6. Literature Update

To account for developments between the initial database searches, a further review was made on 11 August 2023 using the same Boolean search string, inclusions, and exclusions from the initial search. This search identified 15 articles, 7 of which were excluded through title and abstract screening. The full texts of the remaining eight articles were read, and all of these were deemed eligible. Table 3 details the screening process. The findings from these articles are addressed in Section 4.

**Table 3.** Literature update screening process (September 2022–August 2023).

| Database | No. of Articles | No. of Articles After Title and Abstract Review | No. of Articles After Full Text Review |
|---|---|---|---|
| EBSCO—GreenFile | 1 | 1 | 1 |
| Web of Science | 2 | 2 | 2 |
| Scopus | 12 | 5 * | 5 |
| ProQuest | 0 | 0 | 0 |
| Google Scholar | 0 | 0 | 0 |
| TOTAL | 15 | 8 | 8 |

Note: * After removal of 2 articles duplicated in Web of Science search.

### 2.7. Scope and Limitations

The rapid practice review provided a broad overview of the topic from the perspective of the academic literature identified within the academic search indexes and the literature from Google Scholar. The original search parameters specified that eligible articles were to be published within a 25-year period of the search made in September 2022. This timeframe was chosen to include the literature related to the emergence of regenerative

design and development theory and practice within mainstream discourse. The timeframe was extended as described to include the most recent literature prior to publication.

Limitations of this exploratory approach include the potential for an incomplete dataset resulting from the selection criteria. It is also acknowledged that limiting the search to articles written in English or which have an English translation may have resulted in the omission of relevant information.

## 3. Articles Overview—Descriptive Results

Understanding the characteristics of the literature studied in this review helped us to appreciate the scope and depth of the literature that was available, and how this influenced the analysis. Given the focus on practitioners in the built environment sector, the descriptive analysis considered disciplines referred to in the articles. The scale of the built environment considered by the literature was also considered an important factor due to the focus on place-based enquiry.

### 3.1. Journal Discipline Characteristics

The journals containing the articles selected for the study were categorized in terms of their relationships to the built environment sector, sustainability, the environment, ecology and biology domains, environmental management, and finally, social issues (see Table 4). A larger table of additional information regarding the journal names related to each of these categories and the databases for each article (Scopus, Web of Science, ProQuest, and EBSCO) is also provided as Supplementary literature to this paper.

**Table 4.** Journal disciplines from primary databases and google scholar (GS).

| Journal Discipline Categories [1] | Subset 1—Primary Database Articles | Subset 2—GS Articles (2018–2022) | Subset 3—GS Articles Expert | Total No. | Total % |
|---|---|---|---|---|---|
| Built Environment Sector | 9 | 2 | 11 | 22 | 29.3 |
| Sustainability | 2 | 9 | 1 | 12 | 16.0 |
| Ecology, Biology, Enviro. | 5 | 2 | 2 | 9 | 12.0 |
| Environmental Management | 4 | 3 | 3 | 10 | 13.3 |
| Social Issues | 5 | 1 | 0 | 6 | 8.0 |
| Book Chapters | 2 | 6 | 0 | 8 | 10.7 |
| Conference Papers | 3 | 0 | 5 | 8 | 10.7 |
| TOTAL | 30 | 23 | 22 | 75 | 100 |

[1] Note: The Supplementary Materials include a list of the specific journals within each category.

The built environment sector journals represented a range of disciplines, including urban design, landscape architecture, architecture, environmental engineering, sustainability, and ecology. The transdisciplinary nature of these articles demonstrated the importance, acceptance, and on-going dialogue around advancing sustainability and regenerative design practices within the sector.

### 3.2. Spatial Scales of the 'Built Environment'

Most articles focused on city and urban scales, but there was also acknowledgement of the overlapping scales of the built environment, as summarized in Table 5. Cities were variously characterized as having the potential to be: regenerative [42] (p. 225), sustainable [26,27], eco- [43], biophilic [42], resilient [44], or smart and wise [45].

**Table 5.** Built environment scale and key ecological concepts in articles.

| Scale (no. of Articles in Subset) | Key Ecological Concepts in Relation to Scale [Articles] |
|---|---|
| City (14—Subset 1) (13—Subsets 2 and 3) | • Ecosystem services [32,46–53]<br>• Urban metabolism [8,11,42,47–49,54–57]<br>• Regenerative and urban biomimicry [46,58]<br>• Urban ecology [6,11,44,46–48,50,53,54,56,57,59–66] |
| Urban (17—Subset 1) (25—Subsets 2 and 3) | • Urban sustainability [11,28,45,67]<br>• GIS and Landsat mapping [62,63,68–70]<br>• Resilience [11,27,28,44,50,71–76]<br>• Biophilic design framework [42,57,77,78]<br>• Ecosystem services [32,49,51–53,58,59,77]<br>• Ecosystem biomimicry [10,46,53,77,79]<br>• Design process to include ecologists [11,32,53,56,80]<br>• Design process to include Indigenous Knowledges [13,64,81] |
| Neighborhood or Precinct (14—Subset 1) (8—Subset 2&3) | • Community gardens (socio-ecological memory) [59]<br>• Community engagement [13,25,29,34,61,82–86]<br>• Participatory design [12,15,26–28,31,34,43,60,82,84,87–95]<br>• Urban Learning Labs [94]<br>• Adaptive co-management [94]<br>• Traditional Ecological Knowledges and storytelling [13,64] |
| Building (14—Subset 1) (3—Subsets 2 and 3) | • Computer modelling of integrated ecology [96,97]<br>• Biomimicry [46,90,98,99]<br>• Ecological systems thinking [100] |
| Global (Theoretical) (15—Subset 1) (23—Subsets 2 and 3) | • Scale-linking [25,95]<br>• Regenerative sustainability [25,27,33,34,41,75,76,86,93,101]<br>• Regenerative design and development [10,26,30,33,42,60,72,73,85,93,102]<br>• Net positive design [26,34,41,72,83,84,86,93,103]<br>• Ecological wisdom [11,24,45,54,64,66,71,104]<br>• Modernity, post-modernity, reflexive modernity [66]<br>• Sustainable Development Goals [27,28,32,41,43,46,60,82,88]<br>• Ecoliteracy curriculum informed by Indigenous Knowledges [81,105] |

This diverse representation of cities is indicative of the overall shift towards reshaping the built environment to better manage the impacts from multiple and intersecting stressors. Many articles with a city focus examined various strategies aimed at mitigating the ecological impacts from city expansion and urban sprawl. Herrmann et al. (2016) took a different approach by exploring the benefits of integrating nature-based solutions into shrinking cities [61].

At building-level scale, two articles addressed the use of computational fluid dynamic (CFD) modelling to analyse the benefits of adding landscape elements to outdoor spaces. One study was based in a tropical climate [96], while the other was focused on a place with cold winter winds [97]. Another article explored how biomimicry could contribute to sustainable outcomes [98], drawing on well-documented examples of biomimicry applied to the built environment. These examples were analyzed within a framework described by Pedersen Zari (2007) consisting of three levels: (1) the organism, (2) their behavior, and (3) system dynamics [99]. The analysis also considered the five related dimensions consisting of form, materials, construction, processes, and functions.

## 4. Thematic Results and Discussion

This section presents the thematic analysis results and discusses the potential for enhancing and co-creating local ecological knowledge by integrating place-based ecological knowledge into projects. Key vocabulary and terms, relating to the four sub-questions that were explored (i.e., building on Figure 1), are summarized in Figure 6. The figure is discussed briefly, and the key findings are presented in the following subsections.

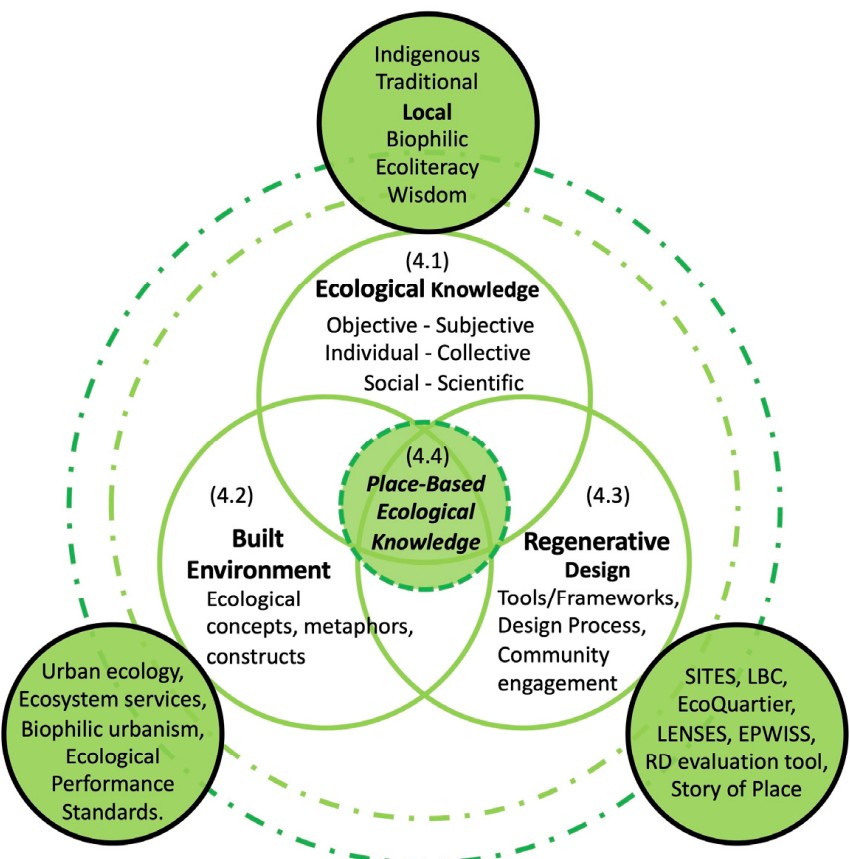

**Figure 6.** Summary of thematic results related to key search terms. The numbers correlate to subsections within the Thematic Results and Discussion section. (Source: Authors).

The three interlaced circles of the Venn diagram relate to the following considerations: (1) Ecological Knowledge, which is characterized by the types of insights and worldviews that can shape the built environment; (2) the Built Environment, which encompasses the ways in which ecological concepts and constructs could be addressed within the built environment sector; and (3) Regenerative Design, which aims to show the ways in which ecological insights may be translated into regenerative design in theory and practice. At the center of the circle, 'Place-Based Ecological Knowledge' refers to the overarching research question that is concerned with how place-based ecological knowledge can be integrated into regenerative design practice. The main concepts identified in the aspects of inquiry are noted in the shaded circles. The dashed concentric circles represent the greater systems within which the research is embedded.

### 4.1. Ecological Knowledge—Definitions and Types

Although humans have long engaged with the study of nature, the concept of ecology is relatively recent, and the field has evolved rapidly over the last few decades [58,63], particularly in relation to its prospects for the built environment [56]. The literature highlights two distinct perspectives in defining ecological knowledge: one rooted in the scientific method, offering objective insights, while the other reflects socially constructed knowledge, characterized as being subjective [71]. Definitions of ecological knowledge from the literature and the contextual perspective from which they are derived are presented in Table 6.

**Table 6.** Ecological knowledge definitions and perspectives.

| Knowledge (No.) | Key Definitions and References | Perspective |
|---|---|---|
| **Ecology** (**Science-**Based Knowledge**) (58)** | • 'The interdisciplinary scientific study of the living conditions of organisms in interaction with each other and with the surroundings, organic as well as inorganic' [34] (p. 115). <br> • '. . . often used to refer to a philosophy, a design strategy, a commercial stance or an ethical position' [44] (p. 144). | Objective, Collective |
| **Ecoliteracy (4)** | • 'The ability to understand the natural systems that make life on earth possible, including understanding the principles of organization of ecological communities (i.e., ecosystems) and using those principles for creating sustainable human communities' [34] (p. 115). | Objective, Individual, Collective |
| **Indigenous Ecological** Knowledges/ **Traditional Ecological** Knowledges **(16)** | • Knowledge that is. . .'sensitive to dynamic, complex spatial and temporal cycles and variations across years [64] (p. 177)' and 'accumulates over time and expands through shared experience and diverse understanding of environmental relationships' [64] (p. 179). <br> • '. . .a cumulative body of knowledge, practice, and belief evolving by adaptive processes and handed down through generations by cultural transmission, about the relationship of living beings (including humans) with one another and with their environment' [71] (p. 1252). | Subjective, Individual, Collective |
| **Local Ecological** Knowledges **(8)** | • '. . .developed communally, over time, in interactions among individuals in the group, cultivating informal shared use of behavioural norms and implicit ways of working and learning together' [59] (p. 262). | Subjective, Individual, Collective |
| **Biophilia (20)** | • Innate and individual knowledge of being a living system and the 'the innately emotional affiliation of human beings to other living organisms' [106] (p. 31). <br> • '. . .premise that because the human mind evolved in the natural world, survival behaviours and responses related to certain organisms, landscapes and natural forms are genetically inherited, and effect the human sense of belonging and wellbeing' [90] (p. 295). | Subjective, Individual |
| **Ecological Wisdom (8)** | • '. . .an understanding of the environment as a holistic system, a spiritual rootedness to place, and affirmation of people's place and responsibilities within it' [64] (p. 535). <br> • Embraced as 'both individual and collective knowledge [65]', cited in [64] (p. 172). <br> • '. . .evidence- based knowledge, tacit and/or explicit, that organizes and evolves from diverse philosophical, cultural, and disciplinary backgrounds across generations, ideally both the process of, and approach to, its acquisition, application should be designed and implemented in a way that is transgenerational, transphilosophical, and transdiciplinary' [65], cited in [66] (p. 95). <br> • 'Wisdom is here considered as the application of a tacit knowledge oriented toward the stability of the urban environment and a city social and economic system' [54] (p. 367). | Subjective, Individual, Collective |
| **Ecological Worldview (36)** | • A way of viewing the world that recognises that humans exist within the interconnected, interrelated, dynamic and complex adaptive living system of Earth [25] (p. 43). <br> • '. . . sees the phenomenal world as constantly regenerated through interactions within systems at all scales and levels of existence (physical, intellectual, emotional, social and spiritual)' [25] (p. 7). | Subjective, Collective, Individual |

In referencing the literature and the emergence of the field, Du Plessis and Brandon (2015) discuss the term 'ecological' as implying 'an understanding of dealing with living

systems and all that comes with such systems, including connections, flows, relationships, interdependence, evolution and consciousness' [75] (p. 6). This appreciation is confirmed by several other researchers, as an umbrella term for knowledge and understanding about ecosystems (living systems) accumulated over time, through experience, observation, shared narratives, and memory [59,64,72,107].

### 4.1.1. 'Scientific' Ecological Knowledge

Most articles present ecological knowledge from a contemporary science perspective [44,83]. Examples of this include urban ecology, ecosystem services, and sustainable rating tools. Scientific ecological knowledge is derived from the scientific method drawing on evidence-based research and experimentation, reflecting the continuity of the mechanistic paradigm within which the modern built environment is founded [83]. This form of knowledge extends to the ecological constructs, metaphors, tools, and frameworks used to inform regenerative design that are described in the following section. A scientific approach to integrating ecology employs a 'measuring' mindset that enables objective knowledge outcomes. The challenges associated with this mindset are evident in critiques of the reductionist approaches used in sustainability tools and frameworks that are seen to limit systems thinking [14,73].

### 4.1.2. 'Socially Constructed' Ecological Knowledges

A growing body of literature recognizes the value of other ways of knowing, subjectively shaped by the lived experiences of individuals and communities through their engagement with specific ecosystems or ecological contexts over extended periods of time [13,64,71,108]. The ecological knowledge gained from this lived experience includes conservation and the sustainable management of biological diversity [4] (p. 29). It reflects dependence of indigenous and traditional communities of working with specific materials and resources within the limits of their given locale. Over time, experiences related to the local ecosystem become embedded within cultural identity and practices constituting ecological knowledge of place [95].

According to several authors, Traditional Ecological Knowledges (TEK) and Indigenous Ecological Knowledges (IEK) offer a compelling aspect of socially constructed ecological knowledge systems [13,64,71,108]. These knowledge systems represent the reciprocal relationship between humans and nature, emphasizing responsible stewardship of nature, or 'caring for country' [13,108]. Reflecting the diversity of First Nations communities, places, and ways of knowing, this paper refers to these socially constructed situated knowledge systems in plural form. The authors acknowledge the emergent potential expressed by Poelina et al. [108], as 'new opportunity and a new respect for Indigenous "Earth wisdom" that has arisen from the impacts of recent climate change and related crises' (p. 3). The authors of this study also advocate for these ancient lineages of knowledge and principles to be respected and the holders of knowledge to be consulted at the earliest stage of projects.

In Australia, Mouritz and Breedon (2022) point to an 'ongoing shift in cultural consciousness' and outline ways that knowledge of country is being brought into built environment projects, specifically through landscape design and management [13] (p. 98). They emphasize the challenges of this awareness are 'fraught with pitfalls of cultural appropriation that necessitates profound acknowledgment and reflection to forge a path of genuine respect and honesty [13] (p. 98).' To design for Country, they stress the importance of genuine engagement and Aboriginal-led methods, concluding that ample time must be allowed to engage deeply and build trust with indigenous knowledge holders and communities [13].

Additionally, Local Ecological Knowledges (LEKs) represent the lived experiences of more contemporary communities. They align with both traditional and Indigenous perspectives in that place-based communities and individuals develop adaptive, dynamic and complex understanding of place that evolves over time [59]. Barthel et al. detail how this knowledge is implanted as social-ecological memory within communities-of-practice

through shared habits, rituals, rules, physical experience, and language [59]. However, urbanization contributes to limiting the development of Local Ecological Knowledges and social-ecological values [62], often resulting from the physical barriers imposed on the experience of natural phenomena [9]. Compounding this loss of experience are the conceptual barriers that arise from the loss of traditional languages that inherently embody ecological knowledge [9,108].

### 4.1.3. Ecological Wisdom

In the literature, ecological wisdom is presented from two perspectives; the first is as an extension of socially constructed ecological knowledge and the second as a synthesis of Traditional Ecological Knowledges and scientific methods. Socially constructed ecological wisdom arises from applied Traditional Ecological Knowledges enacted over time, and accumulated over generations through dynamic learning [64] (p. 179). This interpretation of ecological wisdom is place-based and culturally affirmed, and therefore, attuned and responsive to variable temporal and spatial cycles, thus capable of integrating the unexpected [64].

In the context of the built environment, the concept of ecological wisdom has emerged from a symposium focused on the 'ecological wisdom for urban sustainability' held in China in 2014 [11,45,104]. This evolving concept integrates Chinese intellectual traditions with scientific methods. Landsat and Geographic Information System (GIS) data are used to analyze urban structure and function to inform urban planning decisions [104]. From this perspective, ecological wisdom can be understood as a 'critical synthesis of modernity, post-modernity, and reflexive modernity that embraces the interconnectedness of nature and society and results in a 'beneficial symmetry' in ecological and social outcomes' [66] (p. 97). In this regard, ecological wisdom mirrors the ethical aspirations inherent in regenerative design paradigms aiming to reconcile social-ecological knowledge with scientific knowledge systems.

The discourses around regenerative design paradigms are entangled within those of sustainable design paradigms. Many articles present the differences between these paradigms and the worldview they are informed by. Others see both paradigms existing within a continuum [14,15,66]. In contrast, while messages of scarcity and sacrifice are implied within sustainable discourse [93], regenerative design approaches offer hope by engaging, empowering, and activating communities to participate in and coevolve with the living systems of their place [14,83].

### 4.2. Built Environment Ecological Approaches

The literature discusses a variety approaches to addressing ecology in the built environment sector, with the spectrum of ecological constructs, metaphors, and analogies summarized in Table 7. An ecosystem is an example of an ecological construct in that it is a part of a continuum that allows for manageable study [48] (p. 272).

The body of literature demonstrates the increasing use of the language of ecology contained within ecological constructs (ecosystem services), metaphors (living buildings), and concepts (urban metabolism). Some of this language continues to reflect a mechanistic mindset limiting the potential for a greater transformation. As an example, 'ecosystem services' implies the non-human world is subservient to human needs [27]. From an ecological worldview, a more holistic way of defining these is as 'life support systems' strengthening understanding of their vital necessity for all life and the urgency of repair [57,95]. This serves as an example of the cognitive shift that could influence built environment professionals to better design for regenerative outcomes [109].

**Table 7.** Ecological approaches used by built environment professionals (constructs, metaphors, and concepts).

| Ecological Approach (No. of Articles) | Details and Characteristics of the Construct/Metaphors/Concept |
|---|---|
| Urban Ecology (16) | An evolving interdisciplinary field of research that. . .'seeks to understand the complex and dynamic interactions between socio-economic and natural processes in cities, by considering the whole city as an ecosystem' [11] (p. 11). |
| Ecosystem Services (19) | Benefits the human population derives, directly, or indirectly, from biodiversity and ecosystem functions [94]. Four major categories include: provisioning, regulating, supporting and cultural [1]. Strategies to deliver ecosystem services in urban environments include nature-based solutions and green-blue infrastructure. '. . .a design strategy based on a systematic transfer of scientific ecological knowledge into a built environment context, rather than design based on analogies or metaphors of ecosystems as defined by designers' [51] (p. 56). |
| Ecological Performance Standards (EPS) (2) | Ongoing research championed by Benyus and the Biomimicry Institute. Sustainability goals and metrics based on how a native healthy ecosystem would operate on the site (e.g., quantities of carbon sequestered, water filtered, or air purified). Local ecosystems become models and measures for how a regenerative urban design project in the same location and climate should perform. General EPS Framework: (1) Identify local reference system; (2) quantify ecosystem services to develop EPS metrics; (3) design to meet or exceed EPS metrics; (4) implement and assess [32] (p. 3). |
| Ecosystem Level Biomimicry (9) | Ongoing body of research championed by Pedersen Zari. '. . .flora and fauna of a particular place are studied to find technologies or methods that will fit best to the unique conditions of the site' [10] (p. 34). '. . .strategies based on a transfer of scientific knowledge from ecology rather than design based on the metaphor of ecosystems as defined by designers' [51] (p. 173). |
| Biomimicry (33) | '. . .emulation of strategies seen in the living world as a basis for human design. . . mimicry of an organism, an organism's behaviour or an entire ecosystem in terms of forms, materials, construction methods, processes or functions' [10] (p. 7). |
| Urban Metabolism (12) | '. . .quantification of inputs, outputs, and storage of energy, water nutrients, materials and wastes of urban regions' [47] (p. 34). |
| Biophilic Urbanism (6) | '. . .seeks to use natural elements as purposeful design features in the built environment to provide the benefits of daily exposure to nature' [11] (p. 13). |

Regenerative design theorists and researchers consistently emphasize the vital necessity of making the transition from mechanistic concepts to embrace ecological knowledges [15,34,109]. Several authors acknowledge that this transition is in progress [14,75]. Considering the approaches listed, there is a visible shift in how ecology is influencing design in the built environment, with conservation planning and habitat preservation are recognized to underpin the development of modern ecological science in this sector [63]. Three often-cited ecological constructs are discussed in the following sections, namely 'urban ecology', 'ecosystem services', and 'biophilic urbanism'.

4.2.1. Urban Ecology

Many articles highlighted the scientific basis for ecological knowledge and its contribution to 'urban ecology', which is an emergent discipline in both theory and practice [44,61,63]. Since the 1990s, three distinct paradigms have emerged to address the relationship between the built environment and ecosystems [11,44]. Several articles in the searched literature detailed the importance of these paradigms, as follows:

- 'Ecology "in" the city' was addressed within eight of the articles. This approach applies traditional scientific methods to gain insights on biological or biophysical elements in urban environments, often as a novel comparison to what exists beyond urban boundaries [61]. It is akin to early sustainability practice, stemming from a modernist,

or 'sanitary', view of cities as entities isolated from the surrounding environment and reliant on technical systems to maintain function [11,47,61,66]. In this approach, human activity is seen as a disturbance to natural ecosystems [67]. Strategies that reflect this concept include: design tools that measure ecological elements [62] and community food gardens [59].

- 'Ecology "of" the city' was addressed within 30 of the articles. It expands on and integrates 'Ecology "in" the city' [63] (p. 5) by adopting a social-ecological approach that includes sustainability measures to balance energy and material flows through the built environment [61]. Scientific ecological analogies that align with this approach include urban metabolism [55], circular economy [54], and nature-based solutions [94].
- 'Ecology "for" the city' is informed by the previous two approaches and was addressed within 38 of the papers. This recent iteration of urban ecology 'aims to improve the sustainability and liveability of cities through the application of urban ecological knowledge to the processes of city building in collaboration with stakeholders' [61] (p. 965). The intention is to create ethical, mutually beneficial relationships among living systems by design. Illustrating this approach, Herrmann et al. [61] detail opportunities for partnering with communities to identify nature-based solutions that provide both ecosystem services and social amenity.

Pickett et al. [44] explore the development of this field in detail. They suggest that ecology 'for' the city fosters stewardship through inclusive engagement, highlighting the need for enhanced understanding of urban ecosystems by all participants (p. 5). The evolution in this field parallels that of sustainability paradigms.

### 4.2.2. Ecosystem Services

Ecosystem services were directly addressed by 16 articles across all scales of the built environment. They are largely defined from an anthropocentric perspective as the benefits that natural systems provide to humans [46–48,77]. In urban environments, these benefits include provision of food [59], habitat [77], water treatment [82], and microclimate cooling [96]. Several authors highlight that at the neighborhood scale, linking the delivery of ecosystem services with community engagement and nature-based initiatives creates greater potential for their ongoing success [59,82,87].

Ecosystem services provide both an aspirational goal and performance measures for how cities should function [32,49]. From a biomimicry perspective, the goal is for cities to be 'functionally indistinguishable from the wildland next door' [110] (p. 5). To accomplish this objective, Benyus et al. (2022) propose the use of Ecological Performance Standards (EPS) that use measurable indicators derived from the ecological functions of nearby natural environments. Indicators such as the volume of carbon sequestered and water filtered would then be used to establish place-specific metrics and baseline sustainability goals for the built environment [32]. The ecosystem services and measures addressed by projects need to be prioritized based on local conditions, community needs, and budget constraints [32]. A growing range of tools that rely on Geographic Information Systems (GIS), opensource software models and map data enable ecosystem services to be quantified and ecological indicators to be established [32] (p. 3).

Several articles explored the application of ecosystem service metrics within urban design frameworks to assess sustainable performance and guide planning policy [62,68–70]. Alongside these technological methods, site visits, biodiversity surveys, and researching the local ecological literature are also important to establish qualitative indicators for ecosystem services in the cultural realm [32] (p. 3). Ecosystem level biomimicry frameworks and strategies for integrating ecosystem services into regenerative urban design complement the Ecological Performance Standards. These strategies are evolving within a growing body of research championed by Pedersen Zari [10,46,51–53,77,99,111]. One significant obstacle to implementing these strategies in the built environment is the insufficient ecological knowledge of designers concerning the interdependencies among ecosystem services [46].

Differentiating the measurable ecological functions of complex and adaptive ecosystems to achieve holistic regenerative outcomes poses similar challenges as the categories and lists of sustainability tools. The tendency is to focus on individual elements rather than optimizing synergies among elements. A further challenge of quantifying the benefits of natural processes is that they can be equated to monetary values, which can then be argued to outweigh ecological, social, and cultural values, as has been done before [103,104]. However, built infrastructure that holistically integrates ecosystem services has the potential to act as regenerative nodes 'based on ecological reality rather than human political needs or trends [79] (p. 179).

### 4.2.3. Biophilic Design and Biophilic Urbanism

Twelve articles discussed biophilic design and biophilic urbanism, which is underpinned by the premise that the modern built environment shapes human habitats that, in turn, shape human habitual behaviors [9,81]. A growing body of research on biophilia has identified the negative impacts that disconnection from nature has on human health and well-being, physically, psychologically, and cognitively [11,42,62,75,77,90].

As a remedy, biophilic design seeks to intentionally include nature as a design feature within the built environment to re-establish the human–nature connection [77]. Biophilic design is based on E.O. Wilson's Biophilia Hypothesis, which asserts that humans have an innate tendency to focus on life-like processes resulting from humanity's immersion in natural environments for most of evolutionary history [90,106]. Based on work by Kellert, Browning, Beatley, and others, biophilic design principles and patterns developed since the turn of the century are increasingly being applied within the built environment to deliver beneficial human habitats [14].

Regenerative design tools, such as the Living Building Challenge and Living Community Challenge, promote biophilic design at the building and neighborhood scales. Biophilic urbanism integrates natural elements into the built environment at the urban scale, creating opportunities for humans to tangibly and intangibly connect to natural phenomena [11]. The strategic integration of biophilic design patterns with ecosystems services, referred to by some as biophilic services [11] (p. 9), amplifies the value and potential benefits for non-humans, humans, and the system as a whole [77]. For example, water reclamation gardens provide habitats for non-humans and address the biophilic pattern of the 'presence of water' for humans, whilst also filtering water and offering localized climate regulation [82].

However, as for ecosystem services, biophilic design interventions must be well-adapted to the local context to have ongoing success [82,90]. This imperative is emphasized within ecosystem level biomimicry, which aims to establish mutualistic relationships among the built environment, ecosystems, humans, and living systems [90]. The multiple benefits of biophilic urbanism expand to the social and cultural dimensions of our lives when local communities can be empowered to engage in ongoing project stewardship. The likelihood of this increases when communities are involved in participatory design processes that foster an appreciation of the ecosystem services performed by biophilic design interventions [59,61,82]. Each of the ecological constructs described in this section suggest significant opportunities for future research to explore the strategies for, and benefits of, integrating locally attuned nature-based solutions in conjunction with robust community engagement practices. These research opportunities include analyzing the benefits that accrue from the creation of community partnerships that contribute to meeting the combined aspirations of SDG 11—'Sustainable Cities and Communities' and SDG 17—'Partnerships for the Goals'.

### 4.3. Regenerative Design—In Practice

The literature contains discussions about the disparity between theoretical principles and practical implementation in the built environment sector, arising from low levels of ecoliteracy across society at large, and specifically among professionals in the built environment sector [11,112]. Many built environment professionals, including environ-

mental consultants, who have the responsibility of identifying ecological strategies [56], lack in-depth knowledge of living systems [25]. The benefits of practitioners learning more about the natural world is acknowledged throughout the literature [25,34,57,78]. While scientific knowledge of living systems is greater than ever before [113], regenerative practitioners are called on to increase their ecoliteracy, pattern literacy, and capacity for systems thinking [15,25–27,30,31,34,42,57].

Within discussions about the global potential for regenerative sustainability as a paradigm and in practice, nine articles specifically address the connection with the United Nations Sustainable Development Goals (UN-SDGs), in particular SDG 11—'Sustainable Cities'. Although SDGs are generally acknowledged as an important driver of sustainable practice, the response to them is varied. Tutor (2020) details how nature-based strategies like sustainable water-reclamation gardens have multiple benefits, addressing several SDG, including SDG 11, SDG 6—'Clean Water and Sanitation', UN-SDG 9—'Industry, Innovation and Technology', and SDG 13—'Climate Action' [82] (p. 7). Nature-based and inspired strategies are seen as fundamental to achieving the UN-SDGs, facilitating built environment solutions to positively contribute to local ecosystem services [32,46]. However, addressing individual goals often results in initiatives focused on discrete solutions for individual elements such as energy, water, or vegetation, foregoing the holistic solutions that consider synergies among the elements [41]. This tendency toward reductionism extends from a mechanistic worldview, which emphasizes economic growth, failing to recognize planetary boundaries [43] (p. 83), and promotes human needs above those of the biophysical environment [27] (p. 40). Several authors concur that for the aspirations of the UN-SDGs to be achieved, a new way of viewing the world and humanity's place within it is imperative [27,28,88]. Regenerative design and development is seen as having the potential to achieve this by reconnecting humans with nature, individuals to communities, and communities to each other [27] (p. 41).

Several articles proposed strategies that support adaptive learning about urban ecology and socio-ecological systems. These included integrating ongoing ecological research projects into the design process and as a component of built environment outcomes [48,56]. Felson et al., (2013) detailed a comprehensive approach for including ecologists within design processes [56]. Another approach was explored by Van der Jagt et al. (2019), who proposed that large-scale design projects can function as facilitators of Urban Living Labs [94]. As such, they can generate adaptive social learning processes in conjunction with the formation of learning alliances consisting of communities, organizations, and experts [94]. These reflexive learning strategies emulate feedback loops in living systems that allow for continual improvement and adaptation to place, leading towards coevolution and ecological wisdom.

### 4.3.1. Design Tools and Frameworks

The last thirty years has seen significant growth in the field of 'greening the built environment' [102]. As evidenced throughout the body of literature reviewed, this shift in practice is reflected in the variety of tools and frameworks, both existing and in development, that are used to define and measure sustainability and regenerative design practices. Table 8 summarizes the spectrum of regenerative design tools and frameworks within the body of literature and their approach to ecological knowledge.

**Table 8.** Matrix of regenerative tools and frameworks and their observed approaches to integrating ecology.

| Tool/Framework (No.) | Details | How Ecology is Integrated (Aims) | Knowledge | Refs. |
|---|---|---|---|---|
| **Existing Certification Tools** | | | | |
| SITES (2) | Administered by (US) Green Business Certification Inc. Landscape-focused certification and rating system for sustainable sites. Based on LEED tools (launched in early 2000s). | Aims to create ecologically resilient communities [24]. Supports implementation of nature-based solutions to address a prescribed list of ecosystem services based on the Millennium Ecosystem Assessment (2005) report. | Scientific Objective | [24,104] |
| Living Building Challenge (LBC) and Living Community Challenge (LCC) (14) | Administered by International Living Future Institute (ILFI). A philosophy, certification, and advocacy tool. Mirrors structure of sustainability tools while encouraging regenerative (net positive) outcomes. Categories, referred to as Petals, include Place, Water, Energy, Human Health and Happiness, Materials, Equity, and Beauty (launched 2006). | Aims to restore healthy interrelationship with nature through positive contribution to site ecology by creating ecosystem services, integrating urban agriculture and benefiting the greater ecosystem through habitat exchange. | Metaphor Scientific Objective Experiential Subjective | [27,33,41,78, 83,84] |
| French EcoQuartier (2) | Design framework and labeling program supported by the French government to promote eco-districts. Flexible approach with criteria related to technical, governance, economical, and 'well-being' dynamics. (Launched 2009) | Aims to use ecological and environmental impact studies to inform design, with citizen engagement promoted [41] (p. 3). | Scientific Objective | [12,41] |
| **Proposed tools/frameworks within the body of literature** | | | | |
| Ecological Wisdom Inspired Planning Support System (EWIPSS) (1) | Proposed by Fu et al. (2016) to assess planning scenarios. Ecological Wisdom Index compiled from traditional ecological and socioeconomic indicators and indicators [104] (p. 79). | Aims to use 'Ecological Wisdom Indicators' to relate ecological impacts with human activities (e.g., monetary value of ES, tons of $CO_2$ and CO emissions, structures, and functions of landscape). | Scientific Objective | [104] |
| Regen Concept Framework (1) | Proposed by Svec et al. (2012) to facilitate dialogue on key elements of regenerative practice among leaders in policy, research, practice and local communities; and inspire and support practitioners and community leaders [102]. Consolidates several regenerative frameworks (e.g., LEED, LBC, One Planet) and biomimicry principles. | Aims to encourage systems thinking through a framework that interconnects between nested systems organized within four quadrants: robust and resilient natural systems, high-performing constructed systems, prosperous economic systems, and whole social systems [102] (p. 86). | Metaphor Theoretical | [102] |

**Table 8.** *Cont.*

| Tool/Framework (No.) | Details | How Ecology is Integrated (Aims) | Knowledge | Refs. |
|---|---|---|---|---|
| Decision-Making Framework for Regenerative Precincts | Proposed by Craft et al. (2021) to enable decision-makers to draw on the fundamental principles of regenerative development using a visual guiding framework. | Encourages living systems thinking by understanding key interdependencies, patterns, and place-specific opportunities within the social-ecological system and developing goals to add positive value. | Metaphor Theoretical | [80] |
| Regenerative Design (RD) Evaluation Tool and indicators RCD tool | Proposed by Gibbons et al. (2020) to 'develop greater understanding in inhabitants of a place about how it could function regeneratively as well as foster values, worldviews, and behaviors that support regenerative development [88] (p. 12). | Mimics living systems, guiding communities in perceiving/discovering relationships and patterns that give, have given, or need to be present to bring life and vitality to a place. (p. 33) | Metaphor Theoretical | [29,88] |
| **Frameworks (Including Community Engagement)** | | | | |
| Regenerative Design (and Development) 'Story of Place' (15) | Methodology developed by Regenesis group as part of the regenerative design and development process. 'Story of Place', co-created with community and client, integrates social, ecological, and cultural elements that define unique qualities of place, shape project goals and aspirations, and recognize the potential for the project to contribute positively to place. | Aims to improve pre-design work, to include research into biophysical elements (biological and non-biological) of the local ecosystem, including ecology, topography, hydrology, soil, and climate. These elements are woven into the Story of Place. | Metaphor Theoretical Scientific Experiential-Subjective | [14,15,34,64, 102] |
| Living Environments in Natural, Social, and Economic Systems (LENSES) (19) | Comprehensive holistic framework that offers a process and descriptive metrics to 'create places where natural, social and economic systems can mutually thrive and prosper' [86] (p. 113). Physical layered visual model illustrates interconnections and assists users in seeing, feeling, and understanding whole systems [86]. | A systems approach within the design process considers natural history, ecology [86]. Encourages inclusion of biologists and ecologists within design teams. | Metaphor Theoretical Scientific Experiential Subjective | [33,76,85,86] |

Several articles examined the perceived shortcomings of existing tools/frameworks to develop principles, practices, and new tools/frameworks to support their goals [53,77,80,102,104]. Certification tools, such as LBC, SITES, and EcoQuartier, follow the precedent of early sustainability rating tools, where environmental performance measures are categorized and listed. To some extent, this approach continues to focus on clarifying the role of human pressures upon the environment in a compartmentalized fashion rather than the prospects for restoring and regenerating whole systems [41]. While the performance measures used by these tools aim for net zero or net positive outcomes, there is still the potential to revert to a reductionist approach, dealing with parts of the system rather than the whole [103].

The regenerative frameworks described in Table 8 are contextual and process-oriented, enabling them to be adapted to the unique circumstances and complexity of each project, community, and/or place where they are used. Regenerative design considers site and

place, including the biophysical elements, such as climate, micro-climate, soil, hydrology, and materials. These contextual elements generally inform the design of a project's form, fabric, and relationship to site and surroundings [34].

### 4.3.2. Design Processes

The literature contained an ad-hoc range of engagement, participatory, and co-design processes used to develop place-based socio-ecological information. These processes variously include design charettes, site visits, educational sessions on regenerative design, cultural awareness training, deep listening, and dialogue [76,88,89]. In addressing 'site' (the biophysical elements) and 'place' (the cultural, spiritual, economic, and historic elements), one article highlighted the importance of understanding the causal relationships between site and place, and the greater system [34].

'Story of Place' is an example of a methodology for regenerative community engagement developed by Regenesis Group, pioneering practitioners of regenerative design and development [34]. Building on pre-design information, Story of Place co-creates a narrative that serves to articulate essence of place, align community values and identify ways of collaborating towards ongoing evolution [34] (p. 132). To assist the process of ecological thinking, Mang and Reed (2015) outline a series of questions for practitioners to ask stakeholders about ecological functions and the project's role within the larger system [103] (p. 9). 'Ecology of Place' could be considered as a strand within the Story of Place, however, strategies for holistically obtaining local ecological knowledge through community engagement practices are referenced vaguely.

The exceptions to this relate to biomimicry and biophilic design practices, and community-based research related to ecosystem services. The emergence of biomimicry and biophilic design alongside regenerative design create the potential for new ways of enhancing Local Ecological Knowledges as part of the design process [12,32,90]. Akin to the Story of Place approach, biomimicry practice develops a 'Genius of Place' to identify models and mentors for place-based projects and to develop measures for Ecological Performance Standards [32]. The Living Building Challenge requires a 1-day biophilic design workshop within the design process to identify site specific natural phenomenon and inform design responses that foster (re)connection with nature. Field Environmental Philosophy (FEP) also offers a promising approach for (re)connecting communities with the ecology of their place while also considering the social, cultural, and philosophical aspects [9].

Identifying the stakeholders to involve in regenerative community engagement and participatory design can be challenging, particularly in dense urban environments. A Community of Practice (CoP) is a group of people who regularly interact to share a passion for a concern that they wish to improve [43] (p. 90). A CoP that forms around ecological activities, such as collective gardening, can act as social node of local ecological knowledge gained through collective experience [59]. Identifying the relevant CoP to work through the design process offers rich potential to integrate place-based ecological knowledge into a story of place. Connecting an ecologically focused CoP with the Learning Alliances and Urban Learning Labs suggested by Van der Jagt et al. (2019) can help build regional networks of ecological knowledges, and thus, expand the influence of regenerative design projects [94]. The regenerative design process of deep inquiry into the unique qualities of a place offers a basis for generating collective knowledge and contributing to design solutions that are attuned and responsive to place [33] (p. 51). Purposeful engagement with the communities, who will remain in relationship with the place and ecosystem long after the design teams have left, is fundamental to this outcome [15,34,87,114].

These findings indicate the potential for future research into best practices for community engagement and participatory design processes related to integrating the ecology of place. Research opportunities include, for example, identifying case studies and analyzing the impacts of citizen science as an aspect of participatory design.

### 4.4. Transitioning to 'Place-Based Ecological Knowledge'

The transition from a mechanistic worldview to an ecological worldview is underway within the built environment sector, evidenced by the growing range of interdisciplinary regenerative tools, frameworks, and design approaches. Mang and Reed [30] relate that transitions to new worldviews gain pace as the knowledge constructs that support them manifest across disciplines and fields of endeavor (p. 24). Figure 7 illustrates this ongoing transition, or coevolution, from a mechanistic to an ecological worldview, correlating this with the shift in sustainability paradigms, the development of understanding about relationships between ecology and the built environment, and the progression of modernism as a cultural paradigm.

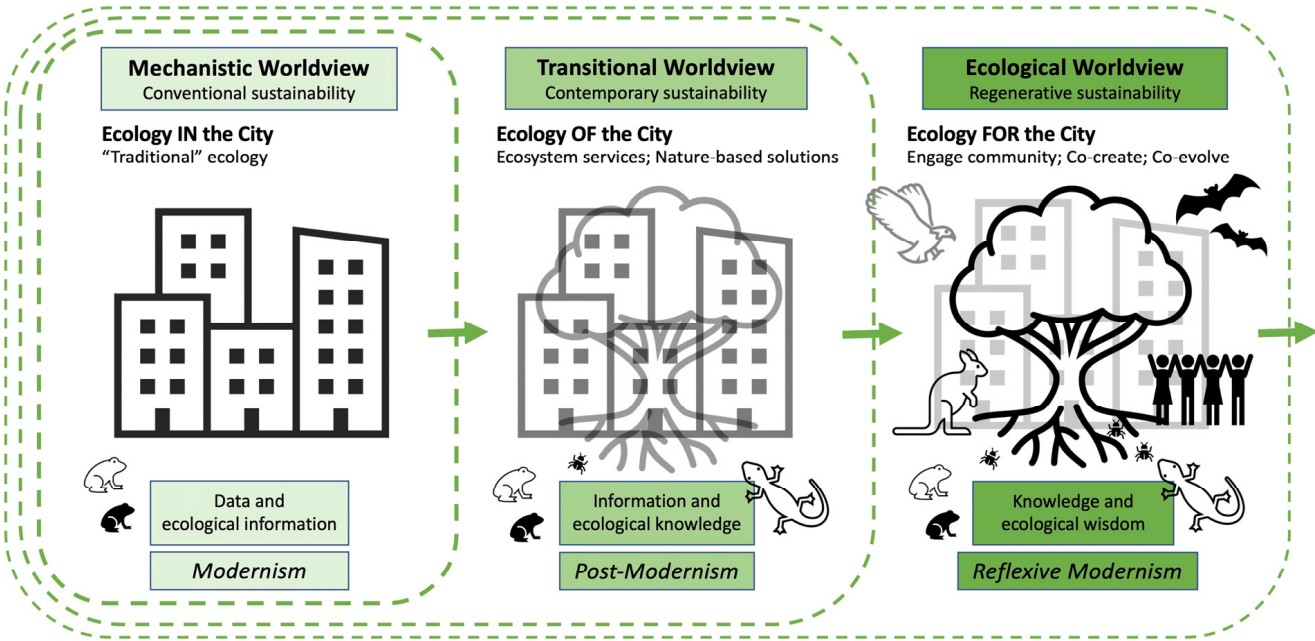

**Figure 7.** Coevolution of sustainability, urban ecology, and social paradigms. (Source: Authors).

This transition is reflected in the field of urban ecology, where the relationship between nature and the built environment has evolved from being separate to inherently intertwined [63]. Each stage of the transition includes and transcends the previous [14] (p. 24), with conventional and contemporary sustainability based on a human-centered mechanistic worldview [28]. From an urban planning perspective, Young (2016) details the shift from modernity to post-modernity to reflexive modernity, promoting ecological wisdom as a guiding principle towards a more equitable synthesis between nature and society based on co-evolution and development [66].

### 4.4.1. Enabling 'People and Place' Coevolution

The findings of this study highlight the presence of knowledge and tools for built environment professionals to draw on and enhance the innate Local Ecological Knowledges held by communities. Processes for achieving participatory regenerative design outcomes are still emergent, with further study needed to explore strategies for enhancing place-based ecological knowledge. The literature includes several key prompts for such study.

Regenerative design paradigms focus on contributing to the evolution of 'whole systems' [75,103]. Gibbon et al., (2020) clarify that a whole system viewpoint consists of complex, nested, and networked hierarchies with interlaced, and therefore, interrelated living systems [29]. Furthermore, smaller scales within the whole drive processes at higher scales, which in turn influence processes within smaller scales [25,50,95]. The implication is that smaller-scale interventions in a specific 'place' can contribute to planetary scale influences [15,115]. A recent systematic review of urban regenerative thinking and practice

found that 'precincts', often conceptualized within place-based frameworks, demonstrate promising opportunities for implementing regenerative design interventions at an optimal scale [72] (p. 340).

A key principle of regenerative design is an on-going partnership between people and their place [101], at all scales [75]. Place-based ecological knowledge fosters a relational ethos that recognizes and respects the interdependence of all living systems [108,116]. Ecological knowledge of place provides a pathway to establishing this ethos and transitioning to the ecologically aligned worldview. Camrass (2020) suggests that inviting people to 'stop, observe and learn from the inherent wisdom that exists in the natural world' has the potential to create a sense of reciprocity that informs human purpose [26] (p. 403).

The ecology of a specific place can serve as an aspiration, a metaphor, and a measure or, as biomimicry paradigms suggest, as a 'model, mentor and measure' for how humanity can fit in on this earth [17,18]. As biological beings, individuals and communities hold innate ecological knowledge of place, whether they are aware of that or not. Social and ecological literacy are key to creating a built environment that contributes to human health and happiness within neighborhood-scale communities [95], (p. 58). Socially constructed place-based ecological knowledge is created by individuals and communities through direct experience of local ecosystems over time.

Rozzi et al. (2012) (p. 10) suggest that most people acquire knowledge of biological and cultural diversity from written materials, digital, and audio–visual resources described by a limited number of languages [9]. This type of literacy differs from the direct experiences of nature that inform indigenous, traditional, and local knowledge systems but points to the potential for engaging with communities 'in place' from wherever the practitioners are around the world. It is through direct experience and deep connection with place and natural phenomenon that biophilic needs are met and the potential for coevolution of people and place exists [114].

4.4.2. Reconciling Knowledges

The themes synthesized from the literature in this rapid review indicate the need for a holistic and inclusive approach to ecological knowledge, to enable place-based ecological knowledge to be integrated into regenerative design and development projects in the built environment. This includes reconciling a persistent disjoint between scientific ecological knowledge and socially constructed knowledges. One regenerative design consultancy documents the process of reconciling conflicting, or paradoxical, forces that arise during the design process by applying the 'Law of Three Forces' [14] (p. 124). This conceptual approach provides a way to view seemingly opposing ideas as being of value, and from that viewpoint, to create something new that harmonizes the value of both [14] (p. 124).

Drawing on this approach, Figure 8 illustrates how the regenerative design process could reconcile the activating force of socially constructed ecological knowledges with the restraining force of scientific ecological knowledge.

Bringing these two ways of knowing together could increase the solution space and the opportunities for people to participate in the evolving system of adaptive learning about place. Benefits could accrue from community building while creating community capacity, agency, and advocacy for people to speak for the ecology of their place. In turn, built environment professionals could learn from communities about the ecology of place, building their own capability and capacity to create well-adapted living buildings that enhance place. Further research to explore these hypotheses is implied.

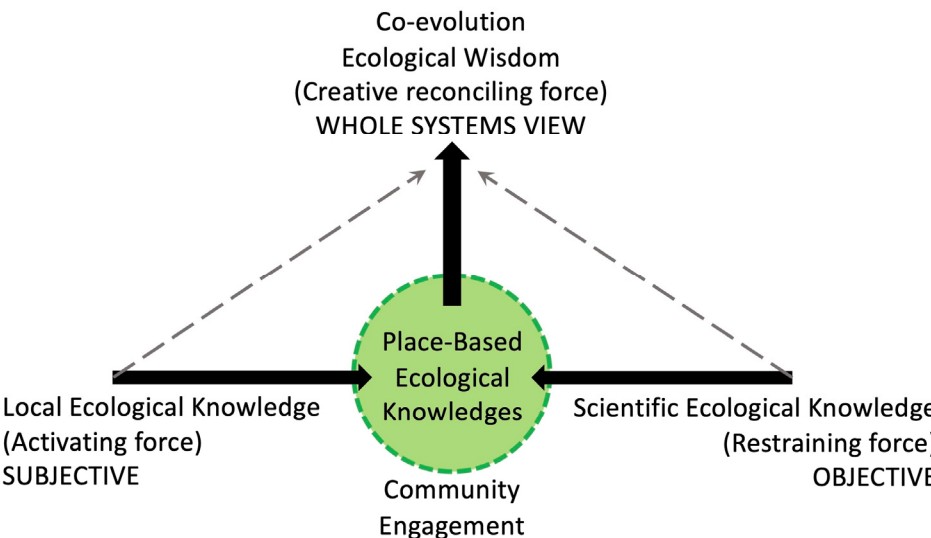

**Figure 8.** The 'Law of Three Forces' applied to reconcile conflicts between scientific and socially constructed knowledge systems to contribute to ecological wisdom. (Source: Authors).

### 4.5. Updated Literature Review

Key findings from an update of the literature to 2023 are presented in Table 9, comprising eight reviewed articles. Column 3 of the table summarizes the key findings of each article in relation to the themes identified in the discussion. The author team reviewed the connections and confirmed no thematic additions to the rapid review findings.

**Table 9.** Summary of updated literature (September 2022—August 2023) and theme alignment to the thematic analysis findings subsections and subsubsections.

| Article Details First Author (Year), Journal [Database] | Title and Key Findings | Themes |
|---|---|---|
| Al-Obaidi, T. (2022), Sustainability [Scopus] [100] | Title: Conceptual Approaches of Health and Wellbeing at the Apartment Building Scale: A Review of Australian Studies<br>• The complex relationships between human health and urban environments necessitate ecological systems thinking. | 3.2, 4.2, 4.2.3 |
| Li (2023), Environmental Education Research [EBSCO] [117] | Title: Developing sense of place through a place-based Indigenous education for sustainable development curriculum.<br>• A strong sense of place positively influences decision-making, enhances cognitive and skills learning, and helps to develop a proactive attitude towards issues faced by community in the future.<br>• Place-based sustainability curriculum is more authentic if co-designed with traditional knowledge holders. | 4.1.2, 4.4 |
| Marshall (2022), Ch3—Design for Regenerative Cities and Landscape [Scopus] [81] | Title: Using Indigenous Knowledge in Climate Resistance Strategies for Future Urban Environments.<br>• Traditional Ecological Knowledges can enhance climate change resilience and adaptation strategies and build local capacity.<br>• Place-based ecological knowledge must be addressed from the outset to have cultural and conceptual meaning. | 4.3, 4.3.2 |

**Table 9.** *Cont.*

| Article Details<br>First Author (Year),<br>Journal [Database] | Title and Key Findings | Themes |
|---|---|---|
| Ou (2022),<br>International Journal of<br>Environmental Research and<br>Public Health<br>[Scopus] [68] | Title: Territorial Pattern Evolution and Its Comprehensive Carrying Capacity Evaluation in the Coastal Area of Beibu Gulf, China.<br><br>• As a scientific basis for informed decision-making at regional scale, ecological carrying capacity (ECC) can be evaluated through measures related to land resources, water resources, and ecological conditions.<br>• Remote sensing tools and mapping are used to determine ecological response to land use changes and effect on ECC. | 4.2, 4.2.1, 4.3.1 |
| Wang, X. (2021)<br>Building Materials for<br>Sustainable and Ecological<br>Environment [Scopus] [69] | Title: A Socio-Ecological Perspective on Green Urbanization and Urban Ecological Intensification<br><br>• 'Ecophronesis' is positioned as ecological practical wisdom, bringing intellectual and ethical considerations together through reflective ecological practices. This concept aligns with the necessity for regenerative paradigms and practice to embrace both inner and outer dimensions of existence to go beyond sustainability suggested within several articles [25–27,29,88]. | 4.1, 4.1.3, 4.4.2 |
| Wu, Q. (2022),<br>Science of the Total<br>Environment<br>[Scopus] [70] | Title: A systematic coupling analysis framework and multi-stage interaction mechanism between urban land use efficiency and ecological carrying capacity (ECC).<br><br>• Coupling analysis examines socio-economic and ecological links via complex modeling and multidisciplinary approaches<br>• An emergent research field that relies heavily on scientific method and is most used often at macro-scale. | 4.2, 4.2.1, 4.3.1 |
| Yates (2023),<br>Urban Science<br>[Web of Science<br>and Scopus] [105] | Title: A Transformative Architectural Pedagogy and Tool for a Time of Converging Crises<br><br>• The Living Systems Well-Being compass, (community-led system change tool) integrates bio-regenerative, nature-positive actions to promote place-specific community wellbeing.<br>• Co-creating regenerative transformations requires adaptive learning processes that are place-based and accepting of diversity. | 4.3, 4.3.1, 4.3.2 |
| Zhao (2023),<br>Buildings<br>[Web of Science<br>and Scopus] [97] | Title: Analysis of Winter Environment Based on CFD Simulation; A Case Study of Feng Shui Layout<br><br>• Feng Shui is framed as Traditional Ecological Knowledge (TEK).<br>• Findings of computer-aided analysis of the impact that Feng Shui practice has on the performance of a courtyard indicated a reduction in windspeed during winter months, creating improved conditions. | 4.1, 4.3.1 |

This update on the original search highlights additional research on the complex interrelationships among humans, the built environment, and ecological systems. The papers also highlight the continued polarity between scientific ecological knowledge and socially constructed ecological knowledge.

## 5. Conclusions

Regenerative design and development theory provides a robust framing for practicing with an ecological worldview. This is considered as fundamental to the built environment functioning as a living system, which makes a positive contribution to other living systems. Few built environment professionals have the requisite knowledge and skills to undertake regenerative design and development, including, for example, understanding how living systems function and how they are attuned and responsive to place.

This rapid practice review focused on the practice of integrating ecological knowledges, considering the literature from the last 26 years (1997–2023). It is clear from the review that

while sustainability paradigms have increased the capability of the built environment sector to deliver more holistic solutions, the solution space continues to be limited by practices and processes defined within a mechanistic context. There is a continued dependence on scientific ecological knowledge to inform design. While this knowledge is still important, socially constructed ecological knowledges offer potential within the scope of a regenerative design project to enhance community connection and encourage ongoing engagement with the ecology of their place.

The rapid practice review highlights opportunities and precedents for addressing the gap between the theory and practice of regenerative design and development. Insights are shared in relation to ecological knowledge types, ways ecology can inform practice through frameworks and processes, and how place-based ecological knowledge contributes to regenerative design and development practice. The findings also highlight the emergence of ecological performance standards as an opportunity to contribute to more robust nature-based measures to inform tools and frameworks.

The findings provide an evidence base for regenerative design approaches to connect communities to the ecological systems of their place, empowering stewardship, and contributing to addressing a number of the United Nations Sustainable Development Goals. This review also establishes a strong foundation for future studies to monitor and evaluate the evolution of regenerative design practice. This includes evaluating emergent strategies to integrate location-specific ecological knowledge into regenerative design practice.

**Supplementary Materials:** The following supporting information can be downloaded at: https://www.mdpi.com/article/10.3390/su151713271/s1, PART A: Database references (ProQuest, Scopus, Web of Science, EBSCO).

**Author Contributions:** Conceptualization, J.T., C.D., K.R., D.H. and S.H.; methodology, J.T., C.D. and K.R.; software, J.T.; validation, J.T., C.D. and K.R.; formal analysis, J.T.; investigation, J.T., C.D., K.R., D.H. and S.H.; data curation, J.T.; writing—original draft preparation, J.T.; writing—review and editing, C.D., K.R., D.H. and S.H.; visualization, J.T., C.D. and D.H.; supervision, C.D., K.R., D.H. and S.H.; project administration, J.T. All authors have read and agreed to the published version of the manuscript.

**Funding:** This research received no external funding. The first author was supported in this research project through a PhD Scholarship from Griffith University.

**Institutional Review Board Statement:** Not applicable.

**Informed Consent Statement:** Not applicable.

**Data Availability Statement:** The data presented in this study are available on request from the corresponding author. The data are not publicly available due to privacy concerns.

**Acknowledgments:** We acknowledge the information specialists at Griffith University: Leanne Stockwell for assistance in determining the search string strategy and Sharron Stapleton for guidance on using the visualization tool, Voyant Tools. We are grateful for input from our colleagues in the weekly Griffith University HDR writing circle, particularly writing expert Karyn Gonano.

**Conflicts of Interest:** The authors declare no conflict of interest.

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
