# Peer review of "Integrating Ecological Knowledge into Regenerative Design: A Rapid Practice Review"

_sustainability, doi:10.3390/su151713271_

Round 1

Reviewer 1 Report

Dear authors,

   Overall, it is a clearly written lit review paper. The research design is sound and rigorous.  The themes identified are insightful.  The findings are clearly presented and discussed.  Some improvements needed are 1). the conclusions section is under developed and needs improvement. 2). the implications section is missing, which should include the theoretical and empirical implications. and 3). the limitations and future studies section should be added.    

Overall, it is a clearly written lit review paper.

Reviewer 2 Report

This manuscript explored current design practices within the built environment for integrating place-based ecological knowledge through a rapid review. The paper is well-structured, containing some interesting and useful information. The authors provide a complete framework on the subject. The results are also well-founded and validated, and the conclusions are well-presented.

- The quality of the study would be further improved if a paragraph were added in the concluding section to discuss the added value to the research community and more in-depth discussions on future implications for policy and practice.

Reviewer 3 Report

sustainability-2481972

Integrating Ecological Knowledge into Regenerative Design: A Rapid Practice Review

Comments to Authors:

It was a matter of joy to read and review this study. Some of the corrections were highlighted. Authors are advised to see and review this correction report well and perform point-to-point corrections as suggested.

1.      Show the timeline within the title and abstract of the reviewed stuff. Differentiate the number of kinds of literature concerning their indexing agencies. Why 206 retrieved articles? Why not 300?

2.      What key methodological approaches were found in the literature to support ecological knowledge in regenerative design? Correct all sections of the manuscript by answering this question.

3.      Paragraphs within the manuscript were found smaller. See the first two of the introduction section. See the limits and maintain the standardized length of paragraphs.

4.      Maintain the hierarchy of citations. Review citation rules. See lines 37, 54, 55, and 58 of the manuscript. Use simple language.

5.      The introduction section is dull. Show facts and figures on the subject matter to verify that your study is need-based. Just arguments can’t support your motive to write this manuscript.

6.      Modify Figure 2 “PRISMA” of the reviewed literature. Its explanations are missing. Show its purpose, features, and scope.

7.      Line 69. Why rapid review method was selected? No relationship is verified from the reviewed literature. Is this method selected with the support of the literature? Why not a systematic review? Justification is needed. Authors have to correct their manuscript.

8.      Section 2.1. Where are the problems? Is it suitable to place this subsection in the manuscript, when the study is review-based, and not focused to mitigate the research problems based on qualitative or quantitative data sets? Justify.

9.      What is the purpose and utility of Figure 1? Are these research questions answered within the manuscript? If yes, what were the ground realities when the study is review-based? If not, then why these questions are placed in Figure 1?

10.  Section 2.3. Write and draw the whole screening process.

11.  Figures 3, 4, and 5 are unclear. Redraw.

12.  Source information may be removed from all Figures and Tables when these are properly cited.

13.  All Figures in the result section should be explained well. This is missing. What purpose these are serving? Are these supporting the aim of the study? Clarify.

14.  Nothing is explained with respect to the objectives of the study in the result section. Correction is needed.

15.  What is the outcome of this study? Who are the Beneficiaries? To answer these specific questions, and related many, insert a new section before the conclusion section.

16.  Rewrite the Conclusion section. Its below average in its current form. See related articles to conclude review studies.

17.  Summarise all sections of the manuscript. New paragraphs may be added to every section of the manuscript.

18.  Complex sentences and redundancies were found in the manuscript. Review the manuscript and correct it.

19.  Focus more on the recent literature of 2023, 2022, 2021, and 2020.

20.  Proofread manuscript.

English needs improvement.

Reviewer 4 Report

Thank you for giving me the opportunity to read your paper. The paper “Integrating Ecological Knowledge into Regenerative Design: A Rapid Practice Review” is interesting for journal readers. But following changes should be done before the consideration to improve the quality of the paper: Author's name and their addresses should not be appeared in manuscript as this is against the prescribed journal instructions. This manuscript failed to present the study debates and failed to discuss the debates. The readability is low and the scientific contribution is short discussed. Research questions, research objectives, research gaps are missing. The author has failed to address their contribution for this research it must be addressed crystal clear in first section. The authors have not discussed discussion and findings in the context of earlier studies and therefore it misses out on important results and its implications. The quality of communication is poor and the authors need to rework on every section to modify the paper to make it close to the benchmarks of the journal. The author failed in most of parts. How this work will be beneficial to future researchers and practice managers of the concerned industries? Better to highlight novelty in the study and define motivations for the research. What is the scope of research in the area? Etc. are also not presented. With these issues the paper should be re-visited and modified. 

must be improved.

Reviewer 5 Report

This is one of the most clean article I have reviewed and the presentation of work is well organized. Only few components may need editing but overall, merit and kudos to the authors.

Round 2

Reviewer 1 Report

 Dear editor,

    After reviewing the revised manuscript, I feel the authors have tried to revise the paper to address the concerns raised.  The quality has been improved to be considered to accept this submission. 

Dear editor,

    After reviewing the revised manuscript, I feel the authors have tried to revise the paper to address the concerns raised.  The quality has been improved to be considered to accept this submission. 

Reviewer 3 Report

Integrating Ecological Knowledge into Regenerative Design: A Rapid Practice Review

sustainability-2481972

Comments to Authors:

1.     Adding timeline to the title. Compliance not made. This would give an open eye to readers about reviewed stuff time. Make certain changes to the title.

2.     Short paragraphs are still found in the manuscript. Compliance not made.

3.     Lines 433-437. Very long and complex sentence.

4.     Avoid using the words “below”, and “above” for Tables and Figures.

5.     Relate your study to SDGs. It is missing in the manuscript. Make corrections in various sections of the manuscript.

6.     Research Contribution and Significance be added as separate subheadings just before Conclusion Section.

7.     Explain all Figures and Tables well. Justify their purpose in the manuscript.

8.     The conclusion and Abstract are usually singly paragraphed.

9.     See Report 1 again and make necessary changes that were oversighted.

Very long sentences were found during review.

Reviewer 4 Report

LR is very weak. Add separate LR section and justify with recent LR.

ok, acceptable.

Round 3

Reviewer 3 Report

Comments to authors:

I must congratulate the authors on their performance and compliance. Meanwhile, it seems like the authors are not agreed with the suggestion to modify the title by giving a timeline to it. I agreed with the explanation given to refute the suggested correction. As it’s the author’s choice to opt or not. However, a few minor modifications are suggested as follows.

1.      The last paragraph should also be included in the remainder of the manuscript.

2.      Why reviewers from academia & industry should review your study? Highlight in different sections of the manuscript.

3.      Add the “scope and limitations” of your study as a separate subheading.

4.      Check similarity index of the manuscript and show it to Editor.

5.      Mostly the conclusion section in the articles is treated as an extended version of the abstract. The Conclusion section should leave the informative knowledge to the reviewers. At the moment, this section of the manuscript looks underrated. My suggestions to improve this section are as follows:

a.      Stick to the scope of the study rather than generalized sentences.

b.       Clarify problems, aims, methods, findings, conclusion, and significant contribution and then link your study to SDGs.

c.       Future directions may also be suggested for advanced research on the subject matter.

English looks alright. However, editor may check for the final time before publications.
